Public domain. CC0 1.0.
# Origins, evolutions, and future directions of Landsat science products for advancing global inland water and coastal ocean observations

Benjamin Page[1] (0000-0002-9871-2406), Christopher J. Crawford[2] (0000-0002-7145-0709), Saeed Arab[3] (0000-0003-1602-8801), Gail Schmidt[3] (0000-0002-9684-8158), Christopher Barnes[2,3] (0000-0002-4608-4364), Danika Wellington[2,3] (0000-0002-2130-0075)

[1]Earth Space Technology Services (ESTS), Contractor to the USGS EROS Center, Sioux Falls, SD, 57198, USA
[2]U.S. Geological Survey (USGS) Earth Resources Observation and Science (EROS) Center, 47914 252th Street, Sioux Falls, SD, 57198, USA
[3]KBR, Inc., Contractor to the USGS EROS Center, Sioux Falls, SD, 57198, USA

*Correspondence to*: Benjamin P. Page (bpage@contractor.usgs.gov)

**Abstract.** In April 2020, U.S. Geological Survey (USGS) Earth Resources Observation and Science (EROS) Center introduced a Level 2 provisional Aquatic Reflectance (AR) product for the Landsat 8 Operational Land Imager (OLI), marking the initial phase in developing a standardized global product for Landsat-derived surface water measurements. The goal of USGS EROS aquatic product research and development is to prepare for an operational processing architecture for Landsat Collection 3 in the late 2020s that will enable use of quality-controlled data for emerging Landsat aquatic science applications. To achieve this, we examine the general performance of the Landsat 8/9 provisional AR product through the Science Algorithms to Operations (SATO) framework alongside quantitative assessment using community driven inland water data records (GLObal Reflectance community dataset for Imaging and optical sensing of Aquatic environments, GLORIA) and radiometric coastal validation platforms (NASA's Ocean Color component of the Aerosol Robotic Network, AERONET-OC). Variability within the validation datasets indicate that the performance of the Landsat 8/9 Provisional AR retrieval is highly context-dependent; errors are minimal in optically simple waters (e.g., clear to moderately turbid coastal waters) but increase considerably in optically complex waters where factors such as elevated levels of turbidity, chlorophyll concentrations, or colored dissolved organic matter (CDOM) dominate the water column. Additionally, this paper examines key algorithmic considerations for atmospheric correction, highlighting factors that influence accuracy, scalability, and computational efficiency necessary for collection processing in the operational Landsat Product Generation System (LPGS). The purpose of this paper is to communicate with aquatic scientists, satellite oceanographers, and the broader Earth observation community on the origins, requirements, challenges, successes, and future objectives for operationalizing global AR data products for Landsat satellite missions.

Public domain. CC0 1.0.





## 1 Introduction

For over a half-century, the Landsat program, a joint agency Earth observing satellite mission between the National
Aeronautics and Space Administration (NASA) and the U.S. Geological Survey (USGS), has provided high-quality global
land and nearshore coastal observations from a suite of medium-resolution imaging satellites (Wulder et al., 2022; Crawford
et al., 2023). Upon the adoption of a collection-based archive processing and management approach in 2016 (Dwyer et al.
2018; Crawford et al. 2023), Landsat data are systematically processed, archived, and distributed by the USGS Earth
Resources Observation and Science (EROS) Center located in Sioux Falls, South Dakota, USA. Through collaboration with
remote sensing subject matter experts and participation from the Landsat Science Team, USGS EROS has developed and
operationalized research-quality Level 1 Top of Atmosphere (TOA) calibrated reflectance and Level 2 atmospherically
corrected surface reflectance and surface temperature products that can be used to map, monitor, assess, and interpret how
Earth's surface has changed as a result of human influence and natural environmental conditions. These open access data
products from Landsat are made publicly available at no cost (Zhu et al. 2019) through the USGS EROS Earth Explorer (EE)
data portal and Machine-to-Machine (M2M) Application Programming Interface (API). USGS also offers direct access to
Landsat data through the Amazon Web Services (AWS) commercial cloud environment in a "Requester Pays" (user incurs
cost for data requests and downloads) bucket configuration (Crawford et al. 2023). This allows researchers, scientists, U.S.
federal and state agencies, and international organizations to utilize Landsat data products for their science applications, and
to facilitate informed land, natural resources, and water management decisions and policies (Wulder et al. 2019).


Landsat Level 2 science product development follows a structured process that involves iterative collaboration between
principal investigator(s) (e.g., a Landsat Science Team member or a U.S. federal agency scientist) and the USGS Landsat
science project to operationalize mature science algorithms. The development phases of this process (discussed in Section 2)
include research, provisional, and operational readiness levels for the generation of science data products. Products that are
considered provisional are available to the public through the EROS Science Processing Architecture (ESPA;
https://espa.cr.usgs.gov) on-demand interface but are actively under USGS internal evaluation and remote sensing
community validation. These algorithms and the resulting product layers may undergo further modifications or
improvements before being considered for operational release.

Although Landsat missions have primarily been designed for observing and monitoring changes in the terrestrial land
environment, Landsat 8 (launched February 2013) and Landsat 9 (launched September 2021) have been used extensively for
aquatic remote sensing applications (Tyler et al., 2022) due to the Operational Land Imager (OLI)'s substantial
improvements in both radiometric data quality and spectral resolution compared to heritage Thematic Mapper (TM) and
Enhanced Thematic Mapper Plus (ETM+) instruments (Roy et al., 2014; Pahlevan et al., 2014; Olmanson et al., 2016).
Compensating for the intervening effects of atmospheric scattering and absorption between the sun, surface, and remote

Public domain. CC0 1.0.



imaging sensor, which vary spatially and temporally, is a necessary processing step to enable reliable monitoring, characterization, and interpretation of the Earth's surface (Vermote et al., 2008; Korkin and Lyapustin, 2023; Thompson et al., 2019; Thompson et al., 2022; Pahlevan et al., 2017). In contrast to brighter terrestrial land surfaces, retrieving atmospherically corrected spectral reflectance information from dark aquatic targets using spaceborne imaging sensors is a
major challenge because the attenuated sunlight reflected from the water is usually only a fraction of the total signal received at the top of atmosphere (Wang, 2010).

In April 2020, U.S. Geological Survey (USGS) Earth Resources Observation and Science (EROS) Center introduced a Level 2 provisional Aquatic Reflectance (AR) product for the Landsat 8 Operational Land Imager (OLI), marking the initial phase
in developing a standardized global product for Landsat-derived surface water measurements. The algorithm to generate AR products for Landsat 8 (and Landsat 9 since launch in September 2021) OLI imagery was adopted from version 8.10.3 of the Level 2 Generation (l2gen) module within the SeaWiFS Data Analysis System (SeaDAS), originally developed by the NASA Ocean Biology Processing Group (OBPG). This software has been the standard processing method for several previous and ongoing NASA ocean color missions like the Coastal Zone Color Scanner (CZCS, 1978–1986), the Medium
Resolution Imaging Spectrometer (MERIS, 2002–2012), the Geostationary Ocean Color Imager (GOCI, 2010–2021), the Moderate Resolution Imaging Spectroradiometer Aqua (MODIS Aqua, 2002–present), and the Visible Infrared Imaging Radiometer Suite (VIIRS, 2011–present) (Mobley et al., 2016). USGS Level 2 provisional AR products have been available to process and download from the USGS ESPA on-demand interface. These products underwent a refresh in 2022 following the release of Landsat Collection 2 and contain Level 2 AR for the visible to near-infrared (VNIR) spectral bands (OLI bands
1–5) (Fig. 1), intermediate Rayleigh-corrected reflectance ($\rho$rc) for the visible to shortwave infrared (VSWIR) spectral bands (OLI bands 1–7), and other supporting data layers. These provisional AR products are intended for immediate, experimental use by the remote sensing community involved in water quality monitoring, seafloor classification, satellite derived bathymetry, and other surface water mapping applications so that community assessment of their suitability can be used to strengthen AR retrieval performance to operational readiness in support of applications requiring high quality measurements.
Water quality surveying groups like the USGS Water Mission Area already rely on Landsat and Sentinel-2 observations to characterize U.S. national waters (Fickas et al., 2023; Stengel et al., 2023; Meyer et al., 2024), emphasizing the need for operationally generated satellite-derived data in enabling comprehensive and consistent water resource management and assessments.

Satellite-derived AR measurements are a critical asset where in situ data are scarce or costly to collect. Feedback from science applications end users ensures that data outputs are both robust and actionable, fostering trust and reliability across scientific, policy, and operational domains. The goal of USGS EROS aquatic product research and development is to enable emerging Landsat aquatic science applications and prepare for an operational processing architecture for Landsat Collection 3 in the late 2020s. The purpose of this paper is to communicate with aquatic scientists, satellite oceanographers, and the

Public domain. CC0 1.0.

broader Earth observation community on the origins, requirements, challenges, successes, and future objectives for
operationalizing global AR data products for Landsat satellite missions.

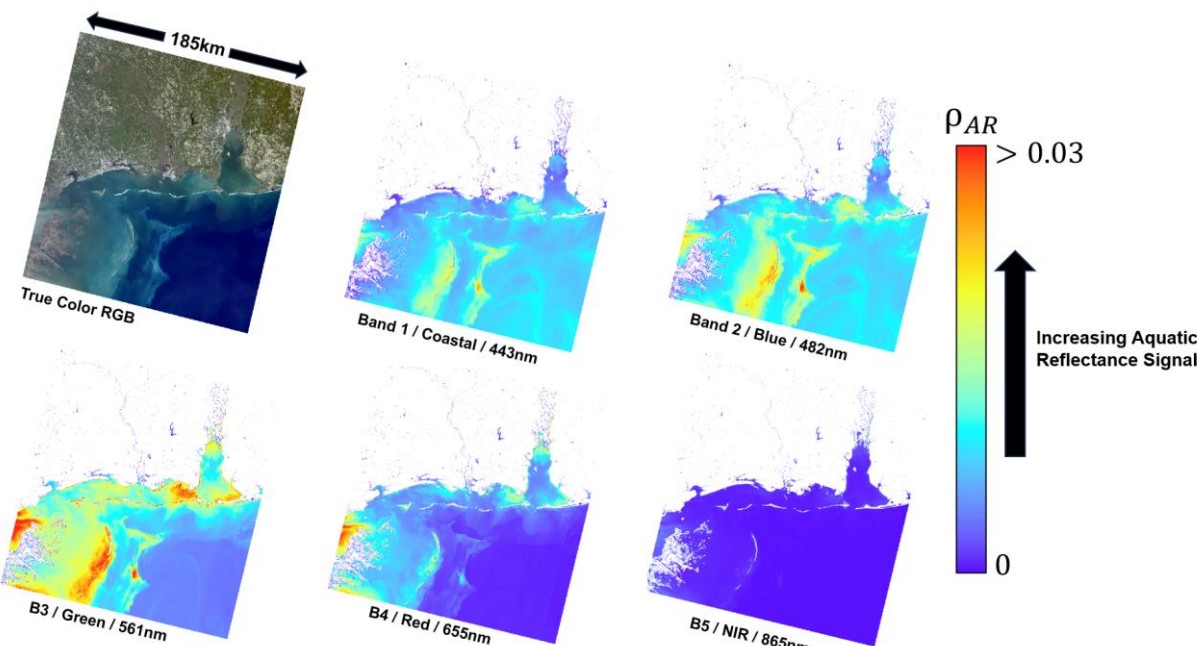

**Figure 1.** Example of the Landsat 8/9 Level 2 Provisional Aquatic Reflectance product over coastal Alabama on November 15[th], 2021.
The Landsat 8/9 Level 2 Provisional AR product package includes AR for the five OLI visible and near infrared (VNIR) bands centered at
443nm (coastal/aerosol), 482nm (blue), 561nm (green), 655nm (red), and 865nm (NIR) for identified water pixels at 30-meter spatial
resolution. *Landsat image courtesy of the U.S. Geological Survey.*

## 2 Landsat provisional aquatic reflectance algorithm description and implementation

Remote sensing reflectance ($R_{rs}$) is defined as the ratio of the spectral distribution of reflected solar radiation upwelling from
just beneath the water surface ($L_w$, W·m−2·sr-1) normalized by the downwelling solar irradiance ($E_d$, W·m−2) in the visible
to near-infrared domain ($\lambda$=400–900 nm, unit: steradian-1) (Lee et al., 1997; Gordon and Wang, 1994; Mobley 1999):

$$R_{rs}(\lambda) = \frac{L_w(\lambda)}{E_d(\lambda)} \ (sr^{-1}) \ ,     \hspace{3cm} (1)$$

$R_{rs}$ is the conventional measurement used in proximal, airborne, and satellite-based remote sensing to quantify the optically
active, biogeochemical constituents (i.e., chlorophyll, total suspended solids, dissolved organic matter) (O'Reilly et al., 1998;
Lee et al., 2001; Mishra and Mishra, 2012; Dogliotti et al., 2015; Concha et al., 2016) and is an essential component for the

Public domain. CC0 1.0.



water quality analysis of lakes (Lehmann et al., 2018; Giardino et al., 2019), long term ocean color monitoring programs (Werdell et al., 2007), benthic mapping practices (Louchard et al., 2003; Dierssen et al., 2010), and optical water type classification for global water bodies (Spyrakos et al., 2018; Bi and Hieronymi, 2024).

SeaDAS, developed and maintained by the NASA's OBPG, is the satellite image preprocessing software for generating aquatic $R_{rs}$ image products for several ocean color missions primarily associated with global monitoring programs for over 25 years (Mobley et al., 2016). Because of this, the open source code for l2gen supports several multispectral (and hyperspectral) Earth Observation missions, including the OLI instruments onboard Landsat 8 and Landsat 9. The adaptation of l2gen processing for use with Landsat OLI data is described by Franz et al. (2015), with additional regional analyses of

the impact of band selection for aerosol estimation provided by Vanhellemont et al. (2014) and Pahlevan et al. (2017).

l2gen within SeaDAS computes the $R_{rs}$ for each band at each identified water pixel from the Level 1 at-sensor radiance $L_t$, which is assumed to be partitioned linearly into distinct physical contributions as shown below:fr

$$L_t(\lambda) = [L_r(\lambda) + L_a(\lambda) + t_{dv}(\lambda)L_{wc}(\lambda) + t_{dv}(\lambda)L_w(\lambda)] \, t_{gv}(\lambda)t_{gs}(\lambda)f_p(\lambda), \qquad (2)$$

$L_r(\lambda)$    = the radiance contribution due to Rayleigh scattering by air molecules

$L_a(\lambda)$    = the contribution due to scattering by aerosols, including multiple scattering interactions with air molecules

$L_{wc}(\lambda)$ = the contribution from water surface whitecaps and foam

$L_w(\lambda)$    = the water-leaving component

$t_{dv}(\lambda)$    = the transmittance of diffuse radiation through the atmosphere in the viewing path from water surface to sensor

$t_{gv}(\lambda)$    = the transmittance loss due to absorbing gases for all upwelling radiation traveling along the sensor view path

$t_{gs}(\lambda)$    = the transmittance to the downwelling solar radiation due to the presence of absorbing gases along the path from Sun to the water surface

$f_p(\lambda)$ = an adjustment for effects of polarization

The l2gen atmospheric correction algorithm retrieves the water-leaving radiance $L_w$ component of interest by estimating and

subtracting the terms on the right-hand side of equation (2) from $L_t$. Of these components, the estimation of the aerosol scattering contribution $L_a$ is generally the most challenging and impactful for the retrieval of $L_w$ (outside of glint-contaminated areas, that is). While the l2gen software accepts a wide variety of processing options for aerosol radiance estimation, the parameterization most commonly used in the operational processing of supported mission data makes use of an iterative bio-optical model to satisfy a fundamental assumption of the algorithmic approach: that near-infrared water-

leaving radiance is either negligible or can be accurately estimated (Bailey et al. 2010). With this assumption, the aerosol

Public domain. CC0 1.0.

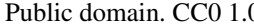

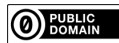

radiance in each band can be estimated via the two-band aerosol selection approach of Gordon and Wang (1994). USGS provisional AR processing uses OLI band 5 (865 nm) and band 6 (1609 nm) as the choice of bands, following the recommendation of Pahlevan et al. (2017). The value of $R_{rs}(\lambda)$ is then computed as:

$$R_{rs}(\lambda) = \frac{L_w(\lambda)}{F_0(\lambda)\, f_s\, cos(\theta_s)\, t\, f_b(\theta) f(\lambda)},$$ (3)

where:

$F_0$ = extraterrestrial solar irradiance (Thuillier et al., 2003)

$F_s$ = adjustment of $F_0$ for variation in Earth-Sun distance

$f_b$ = bidirectional reflectance correction

$f_{(\lambda)}$ = correction for out-of-band response

$t$ = diffuse transmittance

The spectral $R_{rs}$ bands are normalized (multiplied by $\pi$ sr$^{-1}$) to produce dimensionless aquatic reflectance (Franz et al., 2007; Franz et al., 2015; Mobley et al., 2016):

$$Aquatic\ Reflectance\ AR(\lambda) = R_{rs}(\lambda) * \pi,$$ (4)

Additional details, including the full set of processing parameters used in the generation of the provisional AR products, can
be found in USGS documentation (USGS, 2024).

Due to its interoperability, traceability, and availability, the l2gen algorithm in SeaDAS (SeaDAS l2gen 8.10.3) was adopted by the USGS into the EROS's Science Algorithms to Operations (SATO) process in 2018, as a baseline for developing an atmospheric correction pathway for Landsat AR. The SATO  Product Maturity Matrix for USGS Landsat science products is
the formal description of the development process used by USGS EROS to mature algorithms for collection processing in the operational Landsat Product Generation System (LPGS). The purpose of SATO is to enable a smooth transition of researched, developed, and matured science algorithms and prototype executables into a formally developed and maintained LPGS operational environment. The product maturity matrix for provisional Landsat science products is adopted from the National Oceanic and Atmospheric Administration (NOAA) Climate Data Record (CDR) maturity model (Bates & Privette,
2012) and is used as the template to transition select candidate science algorithms through the SATO process (Table 1).

| Maturity Level | | Software Readiness | Metadata | Documentation | Product Validation | Public Access | Utility |
|---|---|---|---|---|---|---|---|
| Research | 1 | Conceptual Development | Little or none | Draft Algorithm Theoretical Basis Document (ATBD); | Little or None | Restricted to a select few | Little or none |

Public domain. CC0 1.0.



| | | | | | | |
|---|---|---|---|---|---|---|
| | | | | paper on algorithm submitted | | |
| | 2 | Significant code changes expected | Research grade | ATBD Version 1+; paper on algorithm reviewed | Minimal | Limited data availability to develop familiarity | Limited or ongoing |
| **Provisional** | 3 | Moderate code changes expected | Research grade, meets international standards | Public ATBD; peer-reviewed publication on algorithm | Uncertainty estimated for select locations / time | Data and source code archived and available; caveats required for use | Assessments have demonstrated positive values |
| | 4 | Some code changes expected | Exists at collection level. Stable. Allows provenance tracking and reproducibility of dataset. Meets international standards for dataset | Public ATBD; Draft Algorithm Description Document (ADD) and Product Guide (PG); peer-reviewed publication on algorithm; paper on product submitted | Uncertainty estimated over widely distributed times / location by multiple investigators; Differences understood | Data and source code archived and publicly available; uncertainty estimates provided; known issues public | May be used in applications; assessments have demonstrated positive value |
| **Operational** | 5 | Minimal code changes expected; stable, portable and reproducible | Complete at collection level. Stable. Allows provenance tracking and reproducibility of dataset. Meets international standards for dataset | Public ATBD, Review version of ADD and PG, peer-reviewed publications on algorithm and product | Consistent uncertainties estimated over most environmental conditions by multiple investigators | Record is archived and available with associated uncertainty estimate; known issues public. Periodically updated | May be used in applications by other investigators; assessments demonstrating positive value |
| | 6 | No code changes expected; Stable and reproducible; portable and operationally efficient | Updated and complete at collection level. Stable. Allows provenance tracking and reproducibility of assessment.. Meets current international standards for dataset | Public ATBD, ADD and PG; Multiple peer-reviewed publications on algorithm and product | Observation strategy designed to reveal systematic errors through independent cross-checks, open inspection, and continuous interrogation; quantified errors | Record is publicly available from Long-Term archive; Regularly updated | Used in published applications; may be used by industry; assessments demonstrating positive value |

**Table 1**. The Science Algorithms to Operations (SATO) Product Maturity Matrix for Landsat science products, adopted and modified from the NOAA Climate Data Record (CDR) maturity model (Bates & Privette, 2012).


The progression and transformation of the product follow a structured procedure, with milestones and responsibilities agreed

Public domain. CC0 1.0.

on between the USGS Landsat science project and the algorithm principal investigator(s). Work is divided into a series of sequential phases, as follows:

**Research Stage (Maturity Levels 1 and 2)**: During this stage, academic researchers and principal investigators lead the process. The product remains publicly restricted until it is published, because significant source code changes are expected. Meanwhile, principal investigators submit peer-reviewed journal articles describing the algorithmic approach.

**Provisional Stage (Maturity Levels 3 and 4)**: Research and development entities, such as USGS EROS, lead and optimize the execution of the algorithm approach. A provisional version of the product becomes publicly available on-demand. Source code modifications continue, and metadata, documentation, and the Algorithm Description Document (ADD) and Product Guide (PG) are published along with the provisional product package. Algorithm uncertainties are estimated, and product limitations are documented.

**Operational Stage (Maturity Levels 5 and 6)**: Operational entities, like the USGS EROS Data Processing and Archive System (DPAS), lead this stage. The algorithm is ported into an operational environment and publicly distributed for operational applications. It is stable, reproducible, and its provenance is recorded in standardized metadata. Peer-reviewed validation methods and published algorithms ensure reliability. Known issues and uncertainties are transparently disclosed.

Throughout a product's provisional lifetime, modifications to its features are expected, although the underlying algorithm to generate the product (e.g., aquatic reflectance) is unchanged. For example, algorithm ingestion into ESPA often involves modifying source code for greater processing efficiency as well as for reproducibility. Science verification at each step is conducted to ensure no anomalies are detected in the data and that any alterations or updates to the source code do not have a direct impact on the algorithm itself. Metadata standards are used to ensure product attributes are an accurate representation of the data, are understandable, and can be referenced. After verification and quality checks, the data product is released through the ESPA on-demand interface for public availability along with documentation and any known caveats published on the USGS product web page. Provisional data products are generated to enable timely scientific use and garner user feedback on quality, algorithm performance, observed uncertainties over diverse geographical regions, and community validation following early adopter feedback. It is the responsibility of USGS EROS to compile this information from the community, work with corresponding research groups, and routinely assess other candidate algorithms with potential principal investigators.

## 3 Key Takeaways

Since their release to the public in 2020, order requests for the Landsat 8/9 Level 2 Provisional AR products from ESPA by the community have now surpassed 90,000 scene downloads as of the end of September 30[th], 2024 (Fig. 2). Maximum downloads were observed during the first year of release (and the re-release, following the availability of Collection 2), followed by downward trends with each passing fiscal year. The release of Landsat 8/9 Provisional AR products allowed the

Public domain. CC0 1.0.

Earth System
Science
Data

opportunity to gain insights from the scientific user community on the quality and accuracy of the products. Examples of
product feedback include research articles and agency reports that evaluate provisional Landsat AR products across a variety
of aquatic scientific applications, including coastal ocean color mapping (Nazeer et al., 2020; Tavora et al., 2023), lake water
quality monitoring (Ogashawara et al., 2020; Niroumand-Jadidi, et al., 2022), and satellite-derived bathymetry (Poppenga &
Danielson, 2021).

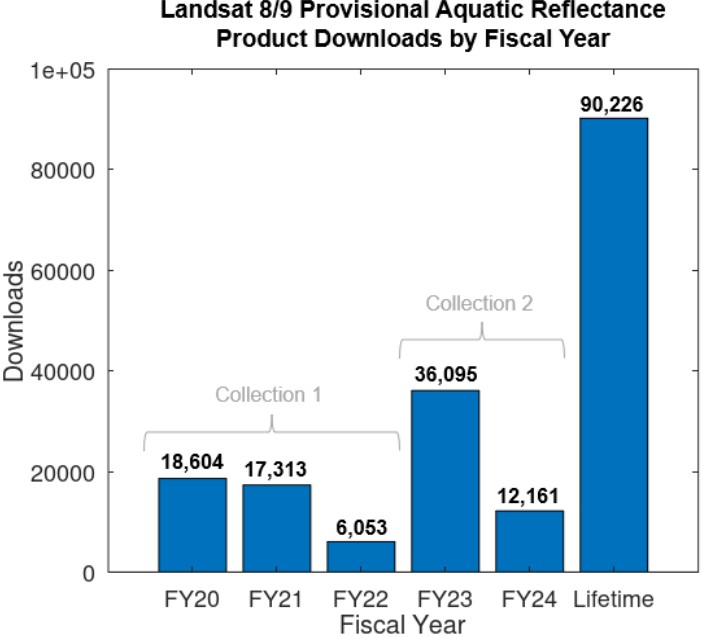


**Figure 2.** Annual download metrics of the Landsat 8/9 provisional AR science products. While not formally part of a Collection
themselves, the AR products have been released using either Collection 1 or Collection 2 input data.

Landsat 8/9 Provisional AR product limitations were recognized by the scientific community concerning (1) the omission of
valid water pixels associated with the l2gen-based land/water delineation and (2) negative AR values generated primarily
over inland and optically complex coastal waters (Pahlevan et al., 2019; Ilori et al., 2019; Ogashawara et al., 2020; Tavora et
al., 2023). While a new water masking approach was developed for the re-release of the provisional products associated with
Collection 2 to mitigate the inconsistencies associated with the l2gen-based land/water delineation, the negative values
resulting from atmospheric correction remain a challenge that has been well documented in the literature across a suite of
ocean color applications (Ruddick et al., 2000; Melin et al., 2011; Bramich et al., 2018; Wei et al., 2018; Kuhn et al., 2019;
Pahlevan et al., 2021). Negative AR, which can significantly affect the accuracy of downstream water quality products, has
been primarily attributed to the challenges of utilizing one or more NIR spectral bands to characterize aerosol path
radiance(s) ($L_a$) over highly turbid or productive, complex case-2 type waters (Bailey et al., 2010; Werdell et al., 2010; Dash

Public domain. CC0 1.0.

et al., 2012; Ibrahim et al., 2019; Wang et al., 2022). In these optically challenging water bodies, the traditional assumption

that water-leaving radiance in the NIR portion of the electromagnetic spectrum is negligible (or effectively estimated by the assumptions of the algorithm) is not valid. Instead, such algorithms may underestimate the substantial water-leaving NIR contribution in highly turbid or productive waters, leading to overestimation of $L_a$ and, consequently, dragging the downstream AR to low and even negative values (Fig. 3). Other challenges faced by SeaDAS (and many other algorithms designed for ocean color) include factors such as the difficulty of treating adjacency effects and the marine bias within its

suite of aerosol models. This issue is intensified for inland freshwater systems, which contain varying amounts of colored dissolved organic matter, suspended sediments, phytoplankton, and surrounding land pixels bordering the entire lake shoreline. Accurate aerosol correction in such environments is crucial for reliable water quality assessments, and addressing these limitations will be decisive for the success of Landsat AR products in future Collections. Increasing user awareness of these issues may explain the observed downward trend in USGS provisional AR product downloads over time. In response,

the provisional product package updates that followed the release of Landsat Collection 2 also augmented the suite of data layers to include AR for the NIR band, per-pixel angle bands, intermediate auxiliary input data and Rayleigh-corrected reflectance products so that users would have supplementary information to further investigate instances when and where full atmospheric correction fails (Table 2). However, these issues must be more fully addressed for the AR product to reach operational maturity. Concurrently, comprehensive aquatic-based atmospheric correction research and applications published

by a variety of authors and institutions have provided alternative approaches that may be better suited to compensate for aerosols in the atmosphere over complex water targets (Steinmetz et al, 2011; Brockmann et al., 2016; Moses et al., 2017; De Keukelaere et al., 2018; Vanhellemont, 2019); consequently, some users could be performing their own processing on Level-1 Landsat data using these alternative approaches rather than relying on the provisional AR products from ESPA.

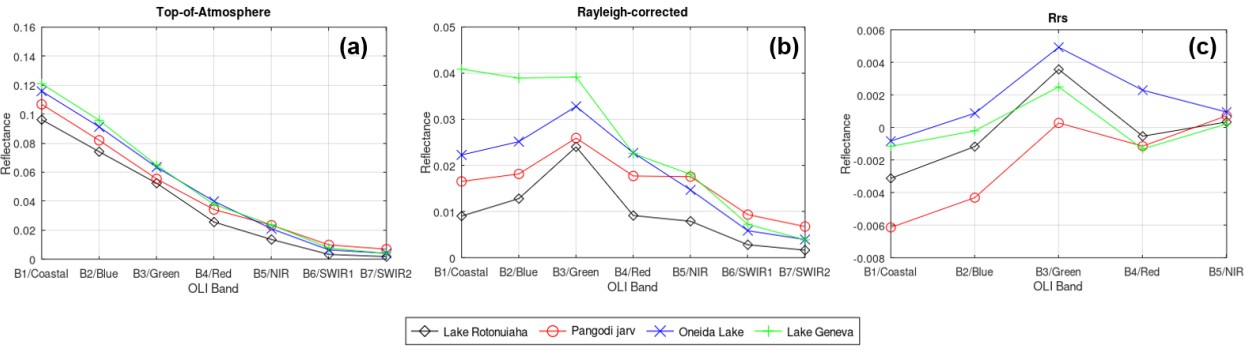


**Figure 3**. Examples of Landsat 8 top-of-atmosphere (TOA) reflectance (left), Rayleigh-corrected reflectance (middle), and Landsat 8 Provisional aquatic remote sensing reflectance ($R_{rs} = AR/\pi$) (right)  for a collection of freshwater bodies, including Lake Rotonuiaha, New Zealand on December 11th, 2017 (LC08_L1TP_072087_20171211_20200902_02_T1), Pangodi järv, Estonia on May 26th, 2018 (LC08_L1TP_187019_20180526_20200901_02_T1),     Oneida     Lake,     New     York,     USA     on     August     30th,     2014

(LC08_L1TP_015030_20140830_20200911_02_T1),     and     Lake     Geneva,     Switzerland     on     April     12th,     2020

Public domain. CC0 1.0.

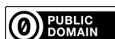



(LC08_L1TP_196027_20200412_20200822_02_T1). Atmospheric interference impacts the spectral profile retrieved by the sensor in low Earth orbit, obscuring key reflectance and absorption features of the optically active constituents in surface waters (a). The Rayleigh correction mitigates single-scattering atmospheric effects, allowing for the retrieval of representative spectral profiles of diverse water targets (b). However, overcorrection of aerosols can lead to negative provisional AR spectra in the VIS bands (c).


| Description | Band Name | Unit |
| --- | --- | --- |
| Aquatic Reflectance Bands 1-4 (VIS) | AR_BAND (1-4) | Unitless |
| Aquatic Reflectance Band 5 (NIR) | AR_BAND5 | Unitless |
| Rayleigh-Corrected Reflectance Bands 1-7 (VSWIR) | RHORC_BAND(1-7) | Unitless |
| Elevation | HEIGHT | Meters |
| Vertical Columnar Ozone ($O_3$) | OZONE | Dobson Unit |
| Water Vapor | WATER_VAPOR | $g/cm^2$ |
| Surface Pressure | PRESSURE | Millibars |
| Wind Speed | WINDSPEED | m/s |
| Tropospheric $NO_2$ | NO2_TROPO | $10^{15}$ molecules/ $cm^2$ |
| Scattering Angle | SCATTANG | Degrees |
| Processing Flags | L2_FLAGS | N/A |
| Water Mask | WATER_MASK | N/A |
| Level 1 Pixel Quality Assessment | QA_PIXEL | Bit Index |
| Level 1 Solar Zenith Angle | SZA | Degrees |
| Level 1 Solar Azimuth Angle | SAA | Degrees |
| Level 1 Viewing Zenith Angle | VZA | Degrees |
| Level 1 Viewing Azimuth Angle | VAA | Degrees |
| Level 2 XML Metadata file | .xml/.MTL | N/A |

**Table 2**. Landsat 8/9 Provisional AR product package contents. Items highlighted were added following the release of Collection 2. Downloads are delivered inside of a .tar file, in a compressed zip file (tar.gz) named in a similar fashion to other Landsat products available from ESPA. Additional specifications and attributes for these files can be found in Section 3 of the Landsat 8/9 Provisional
Aquatic Reflectance Product Guide (USGS, 2025).

# 4 Research Methods

## 4.1 Toward reliable validation of Landsat aquatic reflectance

Public domain. CC0 1.0.



The USGS EROS SATO maturity matrix requires uncertainty estimates of varying sophistication at different product maturity levels. In practice, rigorous estimates of uncertainty are difficult to achieve and assessments of the quality of the product suite instead rely on comparisons of satellite data with in situ measurements. Limitations on the ability to validate the in-development Landsat 8/9 AR products have contributed to these data remaining in the provisional stage. Indeed, finding a collection of reliable validation datasets that represents the full spectrum of optical variability of inland waters

observable by Landsat has been challenging. Previous validation efforts for aquatic based atmospheric correction processors over surface waters in the optical domain have relied heavily on NASA's Ocean Color component of the Aerosol Robotic Network (AERONET-OC) (Wei et al., 2023) and historical field data records from community-driven observations (Pahlevan et al., 2021; Lehmann et al., 2023). Close agreement between satellite and in situ data is widely recognized within the aquatic community as necessary for ensuring the quality of a remote sensing-based product (Ogashawara et al., 2024)


The AERONET-OC Data Display Interface provides access to normalized water-leaving radiances ($nL_w$) collected in various wavebands by platform-based spectroradiometers across a network of coastal and select inland water bodies. These data are frequently used for vicarious calibration and validation exercises for global ocean color missions (Zibordi, et al., 2006; Zibordi et al., 2009). The ongoing radiometric measurements collected from AERONET-OC platforms, using

calibrated CE-318 sun photometers (Johnson et al., 2022), combined with the systematic Landsat 8/9 multispectral acquisitions, provide frequent matchups (near-coincident observations) that allow the scientific community to characterize and validate Landsat AR algorithm outputs in near real time (Mao et al., 2013; Vanhellemont et al., 2014; Bassani et al., 2016; Mannino et al., 2016; Ilori et al., 2019; Xu et al., 2020; Yan et al., 2023; Arena et al., 2024). Preliminary intercomparison exercises between Landsat 8/9 with AERONET $R_{rs}$ (Eq. 1) data have been used to showcase the fidelity of

Landsat to derive AR measurements that are comparable to those of preceding global ocean color missions. However, the locations of the platforms are biased toward moderately turbid (e.g., 0.3 < total suspended solids [TSS, g m-3] < 1.2 & 0.5 < chlorophyll a [Chl-a, mg m-3] < 2.0) coastal and open ocean waters (Pahlevan et al., 2021). The limited number of inland platforms sit on sizeable freshwater bodies within the United States which include Lake Okeechobee, FL (~1,740 km2); Lake Erie, OH (~25,700 km2); and south Green Bay, WI (~1,360 km2) so that freshwater studies can be conducted with

operational ocean color sensors. These inland water bodies experience highly productive seasonal cyanobacterial blooms, so the platforms are essential for understanding the relationships between chlorophyll concentrations and radiometry with respect to satellite observations (Lekki et al., 2019; Moore et al., 2019). However, these freshwater systems do not adequately represent the full spectrum of optical variability of inland waters observed by Landsat across the globe (Pahlevan et al., 2018).


The GLObal Reflectance community dataset for Imaging and optical sensing of Aquatic environments (GLORIA) was released in 2022 (Lehmann et al., 2023). This collection of 7,572 curated proximal hyperspectral remote sensing measurements from 450 different water bodies worldwide was contributed by researchers across 53 institutions. The $R_{rs}$ data

Public domain. CC0 1.0.

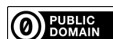



are provided at a resampled 1 nm spectral interval within the 350 to 900 nm wavelength range and are complemented with
several co-located water quality variables (Chl-a, TSS, colored dissolved organic matter [CDOM]) as well as instrumentation
and measurement procedures. Environmental conditions at the time of data acquisition (sky conditions, windspeed,
surrounding land cover, etc.) are also included. The authors have considered the dataset the "de facto state of knowledge" of
in situ coastal and inland aquatic optical diversity, and thus may provide a validation record for the inland waters that is
complementary to freshwater AERONET data. Together, these datasets could help provide insight into the general accuracy
of the Landsat provisional AR products and support the progress of Landsat AR research and development toward the
operational phase.

## 4.2 Validation methodology

Landsat 8/9 OLI acquisitions with accompanying same-day in situ measurements across the combined AERONET-OC and
GLORIA datasets were identified to generate a validation record (Crawford et al., 2025[1,2]). From the 7000+ available
GLORIA $R_{rs}$ measurements between 2013 (launch of Landsat 8) and 2022 (end of GLORIA record), 1,638 matchups ($n_{gloria}$)
coincided with 339 same-day Landsat 8/9 OLI acquisitions with pixels classified by the corresponding pixel quality
assessment layer (QA_PIXEL) as unobscured (no cloud or cloud shadow) water (Fmask 3.3.1, Zhu et al., 2015; Crawford et
al., 2023) (Fig. 4). Matchups resulted in 1,366 water body types representing freshwater lakes, 215 matchups representing
the coastal ocean waters, 17 samples classified as rivers, 31 as estuary, and 9 considered as "other". Corresponding labels of
optical water type for all matchups were subjectively assigned (e.g., "sediment dominated", "chlorophyll dominated",
"clear") by the sample collector as established by the co-located water quality parameter concentration (Chla, TSS, CDOM).

Public domain. CC0 1.0.

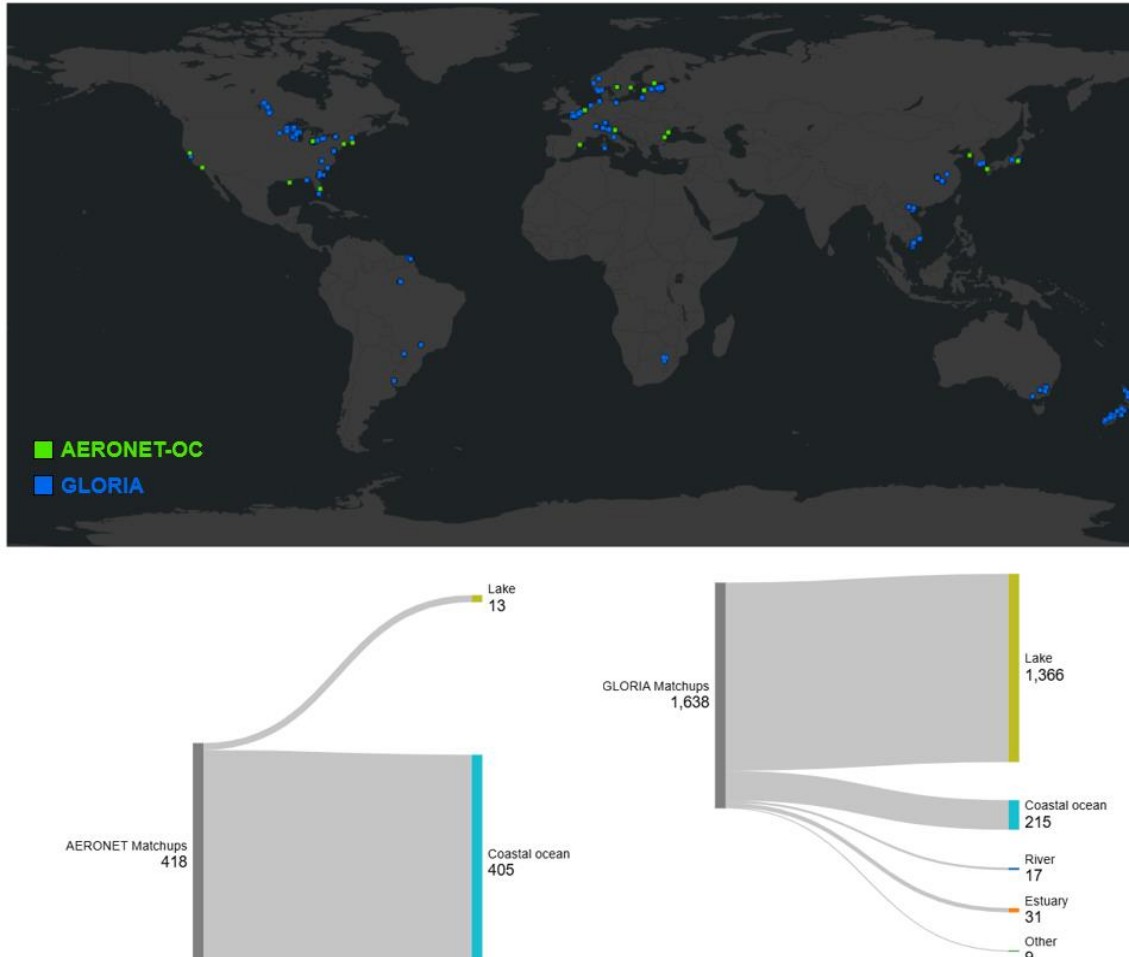

**Figure 4**. Global distribution of the combined AERONET-OC ($n_{aeronet}$=418) and GLORIA ($n_{gloria}$=1,638) matchups with Landsat 8/9 same-day acquisitions (top)- and corresponding classification of water body types (bottom) from each dataset.

In the same manner, 418 AERONET-OC records ($n_{aeronet}$) were found to match up with 412 OLI acquisitions using the same QA_PIXEL cloud filter and temporal window criteria. Level 1.5 AERONET-OC normalized $nL_w$ data were selected to increase the number of available OLI acquisitions per site, despite a potentially lower accuracy than the Level 2 products that may involve a final calibration procedure (Pellegrino et al., 2023). After retrieving $nL_w$ from the AERONET-OC database, $R_{rs}$ was subsequently calculated for each sample:

$$R_{rs}(aeronet) = \frac{nL_{w\_}f/Q(\lambda)}{F_0(\lambda)} (sr^{-1}), \tag{5}$$

where $nL_{w\_}f/Q(\lambda)$ is the Level 1.5 normalized water-leaving radiance, and $F_0$ is the extraterrestrial solar irradiance which has been obtained from the Total and Spectral Solar Irradiance Sensor (Coddington et al., 2021) model and then spectrally convolved with the spectral response function of corresponding Landsat 8/9 OLI sensor.

Public domain. CC0 1.0.

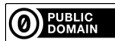



Following the data extraction technique of Pahlevan et al. (2021), average $R_{rs}$ pixel values from a 5x5 window centered on AERONET-OC site were retrieved. To avoid any potential spectral contamination from the platform, the middle 3x3 window of pixels was discarded. For GLORIA matchups, the average pixel values from a 3x3 window centered on the

GLORIA sample location were retrieved. Accuracy assessment was conducted on a per-band basis and employed fundamental statistical metrics often used in ocean color radiometry (Seegers et al., 2018) to evaluate the performance and reliability of the Landsat 8/9 Level 2 Provisional AR products. The Mean Absolute Difference (MAD) was utilized to measure the average magnitude of error between each Landsat 8/9 provisional AR VNIR spectral band and the corresponding band in the merged AERONET-OC and GLORIA in situ validation dataset, providing insight into the overall

accuracy of the geophysical parameter being measured ($R_{rs}$):

$$MAD = \frac{1}{n}\sum_{i=1}^{n}|in\ situ_i - OLI_i|, \tag{6}$$

Additionally, the Mean Absolute Percentage Difference (MAPD%) was calculated to express the relative accuracy as a percentage, enabling comparisons across different datasets:

$$MAPD(\%) = \frac{1}{n}\sum_{i=1}^{n}\left|\frac{in\ situ_i - OLI_i}{in\ situ_i}\right| \ x\ 100\%, \tag{7}$$

The bias metric (β) was incorporated to identify any systematic errors, which determines whether provisional AR products are on average overestimating or underestimating in situ values:

$$\beta = \frac{1}{n}\sum_{i=1}^{n}(OLI_i - in\ situ_i), \tag{8}$$

Finally, the correlation coefficient ($r$) was computed to assess the strength of the linear relationship between in situ $R_{rs}$ and provisional AR products:

$$r = \frac{\sum(OLI_i - \overline{OLI})(in\ situ_i - \overline{in\ situ})}{\sqrt{\sum(OLI_i - \overline{OLI})^2 \sum(in\ situ_i - \overline{in\ situ})^2}}, \tag{9}$$

The Global Climate Observing System (GCOS) scientific community has established threshold (*T), breakthrough (B) and goal (G) targets values of uncertainty for satellite-derived water-leaving reflectance products to be met to ensure that data are useful (GCOS 2025) Here, the observed MAPD(%) between satellite and in situ measurements are used as a stand-in for the GCOS 2-sigma uncertainty metric, which has a threshold requirement of 30%.


## 5.0 Results

Per-band differences between the combined AERONET-OC and GLORIA in situ $R_{rs}$ measurements (n_total=2,056) with the current Landsat 8/9 Level 2 Provisional AR products, on average, did not satisfy the GCOS established threshold *T of 30%. The coastal (B1) and blue (B2) bands returned the highest relative differences of 303.8% (MAD=0.0062 sr-1) and

148.7% (MAD=0.0059 sr-1), respectively, followed by the red (B4) band with a MAPD of 132.4% (MAD=0.0048 sr-1). The green (B3) band had the lowest MAPD of 66% (MAD=0.0062 sr-1) for ground measurements but still did not meet the

Public domain. CC0 1.0.





GCOS established threshold *T of 30%. These immediate observations generally follow the wavelength trends in the l2gen performance for the OLI sensor seen in the aquatic component of the atmospheric correction intercomparison exercise (ACIX-Aqua) (Pahlevan et al., 2021).


The per-band scatter plots shown in Figure (5) provide a closer look into the spread within each of the AERONET-OC and the GLORIA matchup datasets. Most notably, there is a much stronger correlation and less variance between the Landsat 8/9 Provisional AR products with AERONET-OC matchups than with the GLORIA dataset. Secondly, the substantial amount of negative reflectance values generated by the provisional AR products corresponding with GLORIA matchups confirm the difficulties the currently implemented l2gen algorithm has on the inland water processing. Running the accuracy assessment on the two matchup datasets separately also shows substantial differences in performance metrics (MAD, MAPD, β and r) (Fig. 6 / Table 3). For example, the per-band MAPD for the AERONET-OC matchups were considerably smaller than those of the merged dataset, yielding 128.2% (MAD=0.0014 sr-1) for the coastal (B1) band, 45.2% (MAD=0.0012 sr-1) for the blue (B2) band, 35% (MAD=0.0010 sr-1) for the green (B3) band, and 108% (MAD=0.0007 sr-1) for the red (B4) band. In contrast, the per-band MAPD for the GLORIA matchups were larger, yielding 348% (MAD=0.0075 sr-1), 174% (MAD=0.0071 sr-1), 74.4% (MAD=0.0075 sr-1), and 142.2% (MAD=0.0059 sr-1) for the 4 visible (VIS) bands, respectively. Additionally, the correlation coefficient r trend for the AERONET-OC comparison progressively increased toward longer wavelength bands, with a minimum r=0.76 for the coastal aerosol (B1) band and r=0.95 for the red (B4) band. While the r trend did progressively increase with wavelength in the GLORIA comparison, the red (B4) band did not exceed r=0.40, and the coastal aerosol (B1) band was as low as r=0.18. Landsat 8/9 Provisional AR products underestimated both AERONET-OC and GLORIA in situ values on average as determined by negative β but underestimated GLORIA $R_{rs}$ up to six times more than AERONET-OC comparisons. Although none of the spectral bands from the isolated AERONET-OC accuracy assessment met the GCOS *T criteria, the relative error compared to in situ measurements was substantially smaller.


Public domain. CC0 1.0.

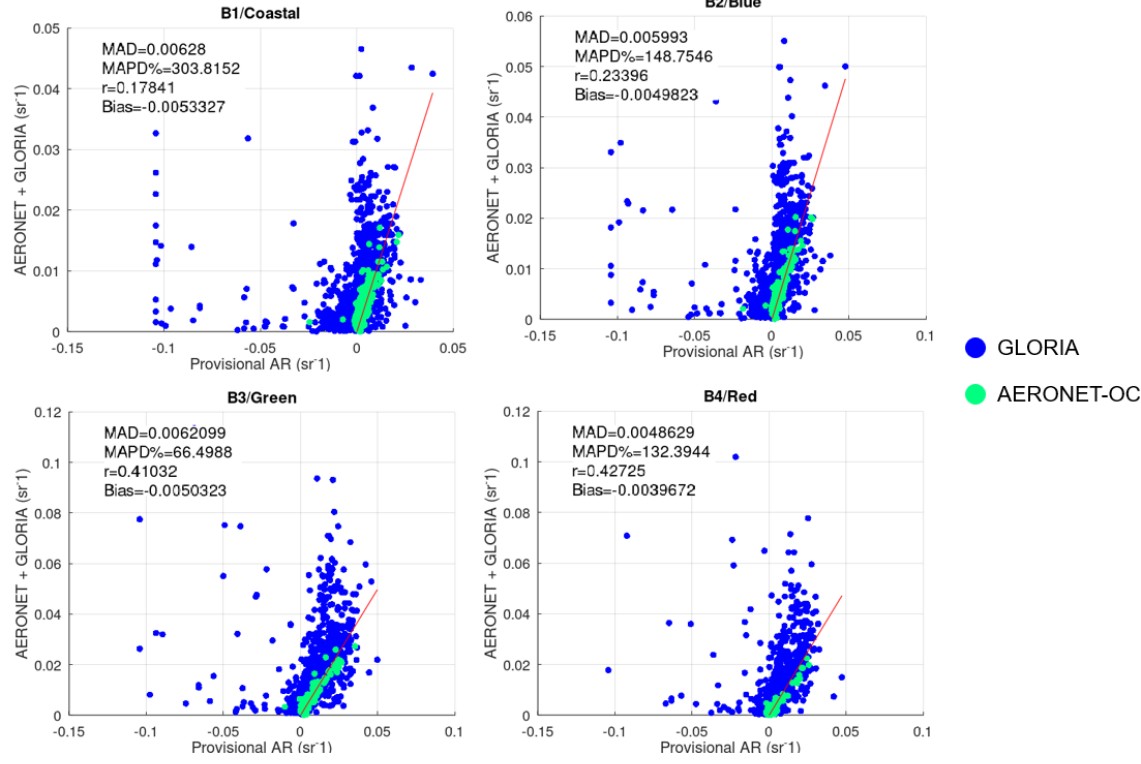

**Figure 5.** Per band scatter plots between Landsat 8/9 Level 2 Provisional AR and the combined AERONET-OC (green) and GLORIA (blue) *in situ* $R_{rs}$ matchups ($n_{total} = 2{,}056$). 1:1 line shown in red.


Public domain. CC0 1.0.

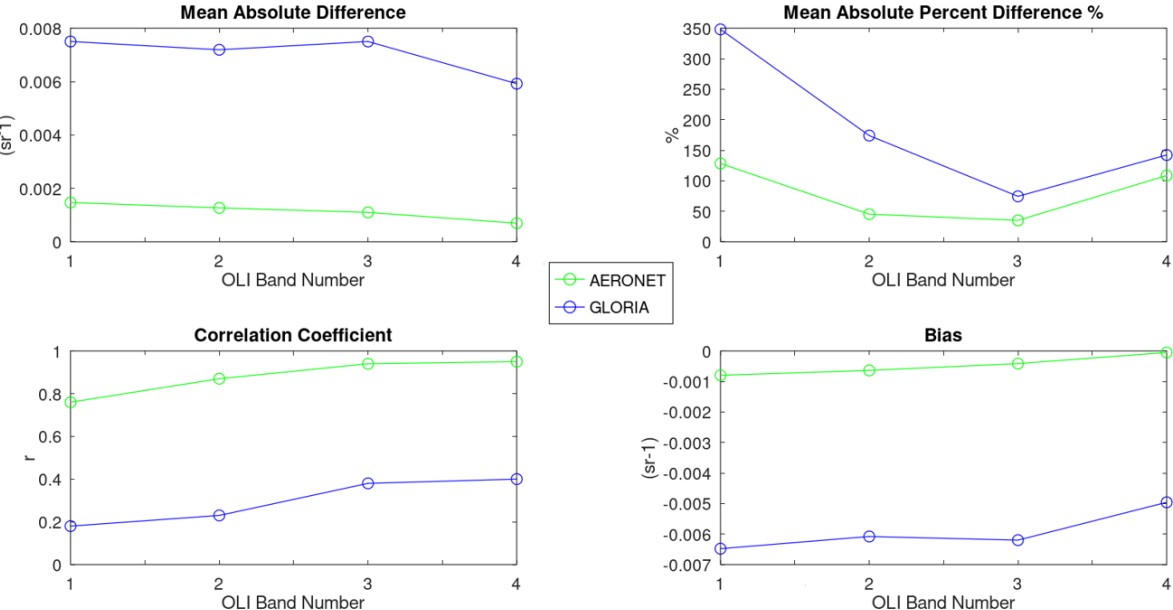

**Figure 6**. Per-band accuracy assessment between isolated AERONET (green) ($n_{aeronet} = 418$) and GLORIA (blue) ($n_{gloria} = 1,638$) $R_{rs}$ with the Landsat 8/9 Level 2 Provisional AR matchups.


| Dataset | OLI Band | MAD (sr⁻¹) | MAPD (%) | r | β |
|---|---|---|---|---|---|
| *AERONET-OC (n=418)* | | | | | |
| | B1/Coastal | 0.0014 | 128.24 | 0.76 | -0.0007 |
| | B2/Blue | 0.0012 | 45.24 | 0.87 | -0.0006 |
| | B3/Green | 0.0010 | 35.07 | 0.94 | -0.0004 |
| | B4/Red | 0.0007 | 108.58 | 0.95 | -0.00005 |
| *GLORIA (n=1,638)* | | | | | |
| | B1/Coastal | 0.0075 | 348.61 | 0.18 | -0.0064 |
| | B2/Blue | 0.0071 | 175.17 | 0.23 | -0.0060 |
| | B3/Green | 0.0075 | 74.42 | 0.38 | -0.0062 |
| | B4/Red | 0.0059 | 142.19 | 0.40 | -0.0049 |
| *COMBINED (n=2,056)* | | | | | |
| | B1/Coastal | 0.0062 | 303.82 | 0.17 | -0.0053 |
| | B2/Blue | 0.0059 | 148.75 | 0.23 | -0.0049 |
| | B3/Green | 0.0062 | 66.49 | 0.41 | -0.0050 |
| | B4/Red | 0.0048 | 132.39 | 0.42 | -0.0039 |

Public domain. CC0 1.0.

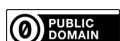



**Table 3**. Tabulated values of the per-band accuracy assessment between each of the set of AERONET-OC ($n_{aeronet} = 418$) and GLORIA ($n_{gloria} = 1,638$) $R_{rs}$ matchups with the Landsat 8/9 Level 2 Provisional AR products.

While the AERONET-OC validation dataset offers a degree of self-consistency due to the use of standardized protocols and instrumentation in retrieving water-leaving radiance from calibrated CE-318 sun photometer measurements, the variability of the GLORIA validation dataset is to be treated with consideration with respect to its consistency and reliability. This variability stems from the diversity of contributors and collection methods (Fig. 7). With data contributions from 34 different organizations, the collection process is subject to differences in protocols, standards, and expertise. Frequent cloud cover, haze, sun glint effects, and unfavourable environmental conditions (e.g., high winds) provide further challenges and diminish

validation opportunities, particularly in low and high latitudes (Radeloff et al., 2024). Although environmental conditions and measurement method were documented for each sample collected (17 different measurement methods total), the inclusion of 22 known radiometer instruments further complicates consistency, because each instrument has varying levels of calibration, accuracy, and uncertainty. Finally, the classification of optical water types adds another layer of subjectivity, because such classifications might not be uniformly applied across all organizations. For example, examining the per-band

MAPD between GLORIA in situ $R_{rs}$ and Landsat 8/9 Level 2 Provisional AR with respect to water type emphasizes how the sensitivity of the validation assessment is strongly influenced by the biogeochemical properties of the surface water. Specifically, the magnitude of the error, reflected in the MAPD, can vary dramatically across different optical water types (Fig. 8). This variability indicates that the performance of the Landsat 8/9 Provisional AR retrieval is highly context-dependent—errors are minimal in optically simple waters (e.g., clear to moderately turbid coastal waters) but increase

considerably in optically complex waters where factors such as elevated levels of turbidity, chlorophyll concentrations, or colored dissolved organic matter (CDOM) dominate the water column.

Public domain. CC0 1.0.

Earth System
Science
Data

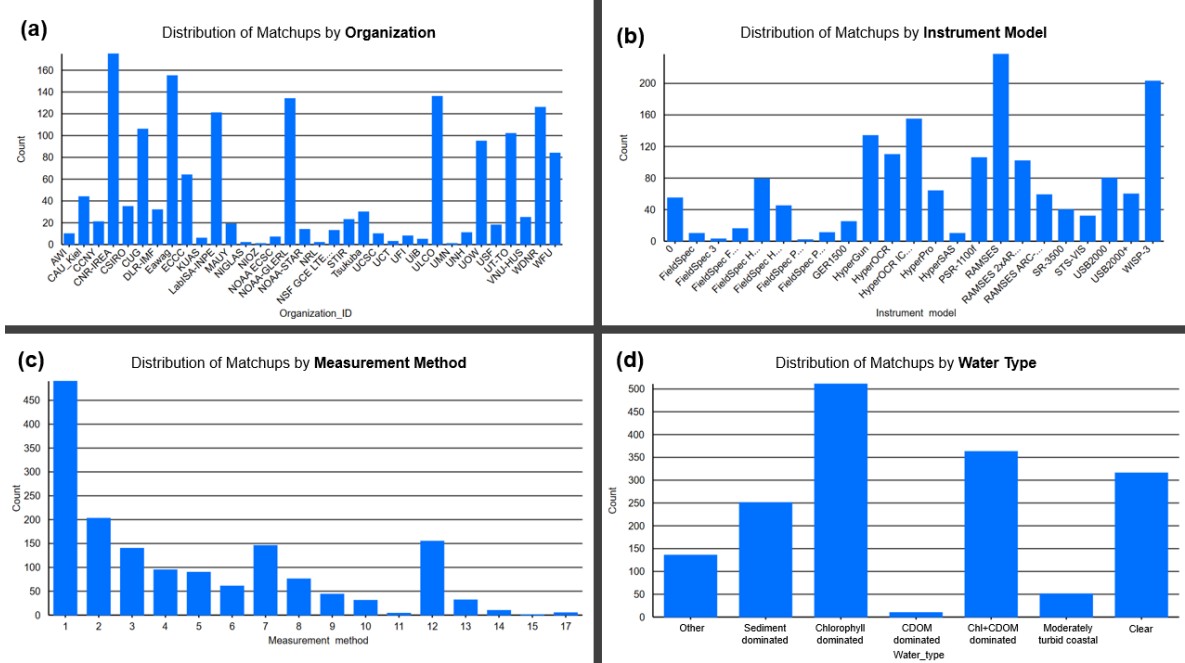

**Figure 7**. Variability of GLORIA matchup sample collection with same-day Landsat 8/9 OLI acquisitions. Matchup sample distribution
include contributions from 34 different organizations (a), 22 known radiometer instruments (b), and 17 different radiometric measurement
methods (c) (refer to Table A.1 in the appendix) across 6 different subjectively classified optical water types (d).

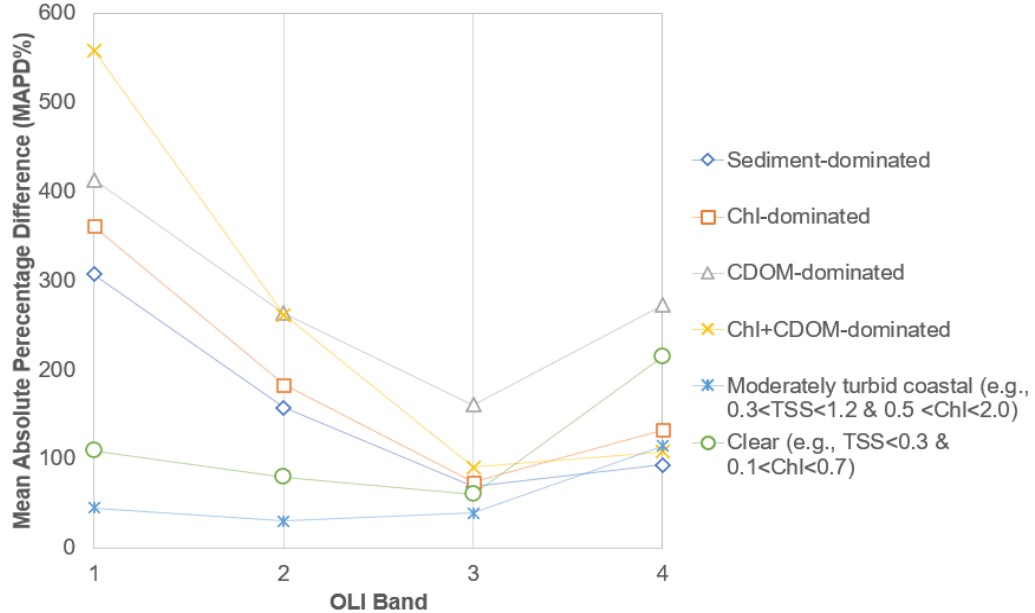

**Figure 8**. Isolated mean absolute percentage differences (MAPD) between GLORIA *in situ* $R_{rs}$ ($n_{gloria} = 1,638$) and Landsat 8/9 Level 2
Provisional AR with respect to water type.

Public domain. CC0 1.0.

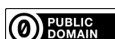



## 6 Discussion

### 6.1 Recent advancements in aquatic reflectance retrieval

Aquatic reflectance represents a particular challenge for the Landsat project, with its emphasis on long-term monitoring, because the performance of heritage Landsat sensors is marginal with respect to the needs of aquatic science (Pahlevan & Schott, 2012; Schott et al., 2016). Improvements in the signal-to-noise ratio (SNR) and radiometric resolution of the Landsat 8 OLI sensor spurred the development of the provisional aquatic reflectance product; however, the results of both the internal evaluation described above and other external evaluations (e.g., Ogashawara et al., 2020) suggest that further re-evaluation of the algorithmic approach and standardizing consistencies for in situ data is warranted. The state of the field of atmospheric correction over water remains fluid, and new approaches and refinements to existing approaches have arisen since USGS began its SATO process for aquatic reflectance. In this section, we briefly review the major directions of research pertaining to atmospheric correction over water.

We broadly classify aquatic reflectance processors based on the major assumptions or characteristics of their approach, as follows: (a) corrections based on a variant of the "black pixel" assumption, (b) spectral ratios and spectral shape matching, (c) machine-learning assisted inversion of forward radiative transfer modelling, and (d) over land atmospheric correction for surface reflectance adapted to additionally retrieve aquatic reflectance.

The "black pixel" approaches to estimating the aerosol contribution are well-known in remote sensing literature and rely on an assumption that water-leaving radiance is negligible/correctable in at least one (if an aerosol model is known or assumed) or two (if an aerosol model is to be selected) bands. For Landsat 8/9, we have already described the implementation of an l2gen-based provisional algorithm, which relies on a pairing of the NIR and SWIR bands to estimate aerosol radiance. This choice arises in part from the lack of a second NIR band on Landsat OLI; the traditional ocean color remote sensing approach involves two bands in the 700–900 nm range (Wang & Gordon, 2018). Other approaches exist that select SWIR bands (Vanhellemont & Ruddick, 2015; He and Chen, 2014) or even a deep blue band (He et al., 2012). A more dynamic approach taken by the "dark spectrum fitting" (DSF) algorithm implemented within the ACOLITE processor allows potentially any band to contribute to the aerosol retrieval (Vanhellemont 2019, Vanhellemont & Ruddick 2018). The key motivation in many of these variants is to address the violation of the core assumption of negligible NIR water-leaving radiance for specific optical water types. Due to the widespread use and high heritage of black pixel-based algorithms, they can often be found within well-maintained software packages with cross-mission support.

Other algorithms rely on assumptions surrounding spectral relationships of the radiometric quantities contributing to the signal. These relationships may be formulated on a theoretical basis, based on the absorptive properties of water, or modeled empirically across a range of water compositions. The bio-optical model that functions as a sub-component of l2gen relies on

Public domain. CC0 1.0.

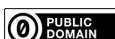

empirically derived relationships across the visible wavelengths to support iterative Rrs(NIR) estimation (Bailey et al., 2010). An approach by Ruddick et al. (2000) relies on the relative invariance of the shape of water-leaving reflectance in the 700–900 nm near-infrared portion of the spectrum to estimate the aerosol contribution over turbid waters. Other approaches (e.g., Singh & Shanmugam, 2014) have been proposed that make use of multiple band ratios and other spectral relationship across multiple wavelengths to disentangle the spectral variability of aerosols. Finally, a more band agnostic approach to atmospheric correction is taken by the POLYMER processor; developed with a focus on addressing sun glint contamination,
it makes use of spectral matching against all available spectral bands (Steinmetz et al. 2011; Steinmetz & Ramon, 2018).

Machine learning algorithms provide a mechanism for more general assumptions on spectral relationships that are internalized by a neural network during the training process. These models are trained on the output of radiative transfer simulations that are parameterized across a range of water constituents, atmospheric conditions, and observational
characteristics. In-situ bio-optical or radiometric databases aid in developing realistic parameterizations. For example, the Case 2 Regional Coast Colour (C2RCC; Brockmann et al., 2016) processor encompasses separate sets of neural nets, each trained over different ranges of optical parameters derived from the NASA bio-Optical Marine Algorithm Data set (NOMAD; Werdell & Bailey, 2005). The Ocean Color – Simultaneous Marine and Aerosol Retrieval Tool (OC-SMART; Fan et al., 2021) is parameterized from MODIS Aqua Level 3 products to estimate reasonable distributions of aerosol and
water optical properties. An approach based on mixture density networks (MDNs) has been implemented in the AQUAVERSE (AQUAtic inVERSion schEme for remote sensing of fresh and coastal waters; Ashapure et al., 2025) framework, although as the time of this publication, this processor is too new to have been included in formal intercomparison exercises.

A final set of approaches involve leveraging terrestrial surface reflectance algorithms to constrain the aerosol properties and generate aquatic reflectance by correcting the over-water surface reflectance for sun and sky glint. This has been demonstrated within the iCOR processor (De Keukelaere et al., 2018), which showed good performance in match-up intercomparisons (Pahlevan et al., 2021). This manner of approach provides a considerable reduction in complexity by reducing the number of algorithms that must be maintained. However, these algorithms rely on scene content that might be
sparse or absent for some over-water footprints; as such, the performance in such areas would depend on the fidelity of the algorithm's internal fallback approach. Other approaches include those that offer a consistent framework that can be applied to retrieve surface or aquatic reflectance (e.g., Thompson et al., 2019).

The differences between the above algorithms predominantly focus on atmospheric characterization, but other radiometric
components have been highlighted within the research community as outstanding concerns. Sun glint and adjacency effects are two such issues. Some atmospheric correction processors include a correction for one or both; however, at the level of algorithm intercomparison exercises, sun glint and adjacency effect components are not typically evaluated separately.

Public domain. CC0 1.0.

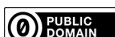

Landsat does not have the anti-sunward tilt that many ocean color sensors use to avoid high glint risk geometries; as such, pixels from certain observations (particularly those acquired at lower latitudes) will suffer from glint contamination. Scattered light from nearby landmasses or clouds provides excess signal to darker water bodies that can interact with algorithms in complex ways (Wu et al., 2024). Providing users with detailed quality information at the pixel level to enable users to filter out potentially problematic data is one mitigating strategy (e.g., CEOS, 2022) but research to better characterize and remove these contributions will further improve data utility.

**6.2 Considerations for Landsat algorithm adoption**

USGS continuously evaluates the state of the field for maturing science algorithms relevant to its Level-2 science product goals. Key criteria that are considered when evaluating external algorithms include (1) a robust presence in the scientific literature, including intercomparison exercises; (2) global applicability across a broad range of environmental and observational conditions; (3) ability to maintain consistency across the Landsat historical record; (4) support for multiple Landsat sensor generations; (5) free, open source algorithm code for which only moderate further development is required; and (6) ability of the code to run at operational scales within reasonable budgetary constraints, after optimization.

Criteria 1–2 are meant to promote algorithms that are well-supported by evidence and garnering interest within the research community. With a few exceptions, the algorithms mentioned in the previous section are found in one of several published algorithm intercomparisons such as the second Atmospheric Correction Intercomparison eXercise (ACIX-II or ACIX-Aqua; Pahlevan et al., 2021) or the report (currently in draft form) by the International Ocean Colour Coordinating Group (IOCCG; Bailey et al., 2024) regarding atmospheric correction over turbid waters. ACIX-Aqua, jointly organized by NASA and ESA, focused on aquatic retrievals over coastal and inland waters for Landsat-8 and Sentinel-2. In this regard it is more directly relevant than the IOCCG (2019) report, for which the evaluations were performed against MODIS Aqua data. Because Landsat Collection processing is meant to support diverse applications, algorithms must be applicable across a broad range of environmental conditions.

The ACIX exercise indicated that in general, the relative performances of aquatic atmospheric correction processors against in situ data from AERONET-OC and a community validation dataset (CVD) depend on optical water type (OWT) to such a degree that a top-performing processor for one OWT was often a low or bottom performer in another, in one or more wavelengths. Pahlevan et al. (2021) suggest that a "fit-for-purpose" solution that reflects the specific downstream needs may be the best supported approach based on the analysis. It is conceivable that a blend of algorithms may offer a compromise solution (e.g., Wang & Shi, 2007; Liu et al., 2019; Joshi & D'Sa, 2020), at the price of a substantial increase in complexity and risk of introducing spatial artifacts. The IOCCG report similarly found that the most turbid OWT disrupted the algorithm rankings substantially, although in other areas the statistical results seemed less competitive than in the ACIX exercise.

Public domain. CC0 1.0.


Criteria 2–4 reflect the need for algorithms that are robust and flexible, yielding results that are consistent through the historical record. Landsat maintains a high degree of consistency in its heritage spectral bands, even if these are supplemented or adjusted in newer missions, with the expectation that heritage bands should result in a long-term time series that appears seamless across satellite generations. Whether Landsat data pre-dating Landsat 8 is deemed of suitable quality

for an operational aquatic reflectance product remains to be determined. However, it is certain that an AR product will be desirable from future Landsat Next acquisitions due to the expanded set of spectral bands that, in addition to enhancing science capabilities, broadens the potential avenues for atmospheric correction (USGS, 2024). This provides an additional challenge as to whether an approach that best leverages these enhanced capabilities would also be compatible with current (or previous) missions or would require a bespoke algorithm. As the capabilities of Landsat satellites continue to expand,

striking a compromise between complexity and maintainability may become a driving consideration.

Criteria 5–6 focus on several factors relating to software maturity, scalability, and open science. Software development is a key contribution that USGS EROS provides during the SATO process but algorithm code maturity within the research phase is an important factor in determining whether to advance an algorithm further in the SATO phases. Processing requirements

are rarely quantified in algorithm comparisons and it is unclear whether comparisons of processing requirements could be quantitatively compared across processors that vary in level of maturity and may have varying potential for further optimization. Nevertheless, processing millions of Landsat observations (encompassing petabytes of data; Crawford et al., 2023) incurs substantial cost.

**Data Availability**

Creator(s): Christopher Crawford, Benjamin Page, Saeed Arab, Gail Schmidt, Chris Barnes, Danika Wellington
Title: Landsat 8-9 Operational Land Imager (OLI) Level 2 Provisional Aquatic Reflectance Products, Collection 2 Validation Subset
Publisher/Repository: U.S. Geological Survey ScienceBase
Persistent Identifier: https://doi.org/10.5066/P14MBBRM
Publication Year: 2025

**Conclusions**

The development of an operational AR product for Landsat, facilitated by SeaDAS open-source code, provided a global AR processing capability for the Landsat user community. l2gen within SeaDAS has been the flagship processor for generating AR products for Landsat 8 and Landsat 9 OLI data, it may not be the most optimal solution as a single global processor for

Public domain. CC0 1.0.



current, heritage (Landsat 4/5 TM Landsat 7 ETM+), and upcoming Landsat missions (Landsat Next) in terms of suitability for emerging science needs that require analysis ready data for both inland and coastal water quality mapping applications. The Landsat 8/9 Provisional AR performance has shown promising results in the coastal regions, but its reflectance retrieval limitations for inland waters must be acknowledged. These limitations include challenges related to atmospheric correction processing accuracy and consistency across optically and geographically diverse water conditions. Addressing these

limitations will be critical for the success of Landsat AR products in future Collections. Until validation campaigns are conducted on a routine basis, the combined GLORIA and AERONET-OC datasets offer a substantial validation pathway toward operational readiness, ensuring the reliability and accuracy of Landsat's aquatic observations for both freshwater and marine ecosystems. The USGS Landsat science project approach for Landsat AR algorithm research and development recognizes the importance of the SATO process and collaboration with established aquatic principal investigators. Promoting

and maintaining success criteria for a global Landsat Collection 3 AR product while remaining aware of evolving mission specifications for Landsat Next is essential. Key criteria include maintaining consistency across spatial and temporal domains, ensuring interoperability with similar products from other medium resolution multispectral and imaging spectroscopy missions (e.g., Sentinel-2, Environmental Mapping and Analysis Program [EnMAP], Surface Biology and Geology [SBG], Copernicus Hyperspectral Imaging Mission for the Environment [CHIME]) (Pinnel et al., 2024; Alvarez et

al., 2022; Dierssen et al., 2021), and balancing the trade-offs necessary to achieve optimal performance in varying atmospheric and optical water conditions. Looking ahead, the next research steps in preparing for Landsat Collection 3 AR development involves undertaking open science algorithm intercomparisons and quantitative validation that considers heritage missions and Landsat Next science readiness simultaneously. With an expanded set of spectral bands targeting aquatic science applications (e.g. 620nm and 705nm center wavelengths) along with higher spatial resolution (10m, 20m and

30m pixels), Landsat Next is expected to provide enhanced capabilities for water quality monitoring and management at a higher temporal frequency. These efforts will provide a foundation for more comprehensive and reliable AR products, ultimately contributing to enhanced understanding and management of aquatic environments globally.

**Appendix A**

| GLORIA Measurement Method Reference Number | Protocol Used for Radiometric Measurements | |
|---|---|---|
| 1 | Sequential Lt, Lsky, and Es via a plaque on MP* | |
| 2 | Simultaneous Lt, Lsky, and Es on MP* | |
| 3 | Lu(0-) and Es on pole connected to a spectrometer via fiber optics MP* or water edge | from |

Public domain. CC0 1.0.

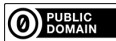



| | |
|---|---|
| 4 | Lw(0+) and Es afloat away from MP* |
| 5 | Lu(0-) afloat away from MP*, Es on MP* |
| 6 | Lt, Lsky, and Es on MP* |
| 7 | Lt, Lsky, and Es on a frame deployed on MP* |
| 8 | Lu(0-) and Ed(0-) in-water profiling from MP*, Es on MP* |
| 9 | Lu(0-) and Ed(z) units on a depth adjustable bar (measurements at -0.21 and -0.67m) on a frame afloat away from MP*, Ed unit lifted above water surface for Es |
| 10 | Lu(0-) and Ed(0-) from winch on MP*, Es on MP* |
| 11 | Lt and Es on pole from water edge |
| 12 | Lu(0-) and Ed(0-) autonomous in-water profiling from a fixed platform |
| 13 | Sequential Lt and Es via a plaque, mounted on gimbal stabilized pole from MP* |
| 14 | Lu(0-) (and Ed(0-) only for depth information) from in-water profiling from MP*, Es recorded simultaneously from same MP* very close to profiler deployment |
| 15 | Lt, Lsky, Es, combined with one Lu unit (aperture at -0.05 to -0.10m) placed on pole |
| 16 | Sequential Lu(0-) and Es via a plaque, both measurements using an optical fiber to a black masked perspex tube |
| 17 | Lu(0-) and Ed(z) units on a floating frame (measurements at -0.4 m (Lu) and -0.1 m (Ed)) drifting 10m away from vessel |

**Table A.1**. Reference table for Fig (7). Brief descriptions of the 17 measurement methods used by each organization that contributed to the GLORIA dataset. For a more detailed definition for each of the protocols, please see Lehmann et al., 2023.

**Author Contribution**

BP led the manuscript. CC oversees the Landsat science production at EROS and provided USGS guidance and expectations. GS was responsible for the implementation of the provisional aquatic reflectance algorithm into the EROS Science Processing Architecture (ESPA) domain. SA assisted with data extraction and processing. CB provided the insight into the Science Algorithms to Operations (SATO) process. DW provided scientific subject matter experience and assisted with the writing process.

**Competing Interests**

The authors declare that they have no conflict of interest.

**Disclaimer**

Any use of trade, firm, or product names is for descriptive purposes only and does not imply endorsement by the U.S. Government. ESTS and KBR, Inc. performed work under contract number 140G0121D001 as part of the USGS Land
Satellite Data Systems Research and Development project.

**Acknowledgements**

We would like to thank USGS colleagues Dr. Keith Loftin of the Kansas Water Science Center and Dr. Victoria Stengel of the Geology, Energy Minerals and Science Center in Texas for their thorough and insightful comments and interpretations of Landsat AR research and developments with respect to the water science domain.

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
