# Peer review of "Origins, evolutions, and future directions of Landsat science products for advancing global inland water and coastal ocean observations"

_Earth System Science Data, 2025_

## Referee Comment (RC1)

**Review of the manuscript "Origins, evolutions, and future directions of Landsat science products for advancing global inland water and coastal ocean observations", by Page et al.**

**General comments**

I am quite confused about this paper. According to the authors, "The purpose of this paper is to communicate with aquatic scientists, satellite oceanographers, and the broader Earth observation community on the origins, requirements, challenges, successes, and future objectives for operationalizing global AR data products for Landsat satellite missions." However, I thought ESSD published papers introducing new datasets. The Landsat Level 2 provisional Aquatic Reflectance (AR) product has been available since 2020. I leave this to editorial decision, but I am not sure whether this journal is the right place for such a paper, which looks more like a review.

**Specific comments**

On Landsat 8/9 validation with in situ Rrs, I will be clear here. The gold standard for such a task is AERONET-OC data. AERONET-OC is the most precise, carefully calibrated data at disposal, with precisely characterized uncertainties, with all instruments within the network following the same procedures, software and calibration laboratories. Instead, GLORIA is pure noise, as already showed in Wei et al. (2025; Fig. 2), and once more shown in Figure 5 of this paper. GLORIA cannot tell anything about the satellite data being compared to, because the uncertainties in the in situ data are, most importantly, unknown. It is merged numbers from a diversity of research groups, most of which do not quite follow best practices on above-water radiometry, so to speak. The authors shall trust me, as I know virtually everybody who contributed to GLORIA. Regarding this, the sentence "Until validation campaigns are conducted on a routine basis, the combined GLORIA and AERONET-OC datasets offer a substantial validation pathway toward operational readiness" shall be removed. In summary, I encourage the authors to remove the comparison with GLORIA or leave them as a demonstration of its unsuitability for satellite validation.

Although this may be a bit off-topic, it is of utmost importance that USGS works in the reprocessing of Landsat 5/7 for aquatic applications and, eventually, the release of a harmonized time series. That would accomplish the main mission of Landsat, which is the development of long-term series. I am aware of the difficulties due to the S/N, band configurations, etc., but any efforts to overcome any of these difficulties would be very impactful.

Line 36: what the authors call "mid-resolution" we actually call "high-resolution". Mid-resolution for us are sensors like MODIS or OLCI.

Line 60: "terrestrial land" sounds redundant.

Lines 90-91: "characterize" sounds vague. What is being done, precisely?

Line 153: is the f_b coefficient the "bidirectional reflectance correction"? I am not sure what is being discussed there, but the bidirectional reflectance correction is a complex process in the

aquatic environment, that involves a bio-optical model, IOP retrieval and some look-up-table indexing. See literature on the matter, especially recent articles, and clarify this concept.

Line 234: "difficulty of treating adjacency effects" Does it mean that l2gen treats the adjacency effect in some way? In this respect, please see a recent contribution by Castagna and Vanhellemont.

Line 277: "community-driven" is more like "community-made"

Lines 287-288: "allow the scientific community to characterize and validate Landsat AR algorithm outputs in near real time". I do not things so, because the AERONET-OC data is a reprocessed product, released some time after the actual measurement.

Section 4.2 "Validation methodology". I have to warn about the satellite vs. in situ data matchup time window. Same day is not close enough. That violates community agreed practices, see Concha et al. A few hours will be already too much, as the coasts are dynamic areas and often tidal.

Equation (5): "$nLw\_f/Q(\lambda)$" that is quite a cumbersome notation. Please choose a better one. By that way, it actually means that the normalized water leaving radiance was corrected for bidirectional effects using Morel's method. Here, there are two issues:

(1) Morel is a method conceptually developed for open oceanic waters. For coastal and inland waters, other methods should be used. See recent works.
(2) If AERONET-OC data is corrected for bidirectional effects and Landsat data is not, as I suspect, that is an inconsistency. Both datasets shall be corrected, and using the same method.

**References**

Castagna, A., & Vanhellemont, Q. (2025). A generalized physics-based correction for adjacency effects. *Applied Optics*, *64*(10), 2719-2743.

Concha, J. A., Bracaglia, M., & Brando, V. E. (2021). Assessing the influence of different validation protocols on Ocean Colour match-up analyses. *Remote Sensing of Environment*, *259*, 112415.

D'Alimonte, D., Kajiyama, T., Pitarch, J., Brando, V. E., Talone, M., Mazeran, C., ... & Gossn, J. I. (2025). Comparison of correction methods for bidirectional effects in ocean colour remote sensing. *Remote Sensing of Environment*, *321*, 114606.

Morel, A., Antoine, D., & Gentili, B. (2002). Bidirectional reflectance of oceanic waters: accounting for Raman emission and varying particle scattering phase function. *Applied Optics*, *41*(30), 6289-6306.

Wei, J., Wang, M., Jiang, L., Lee, Z., Kirby, R., Mikelsons, K., & Lin, G. (2025). Satellite observations of water transparency from VIIRS in global aquatic ecosystems. *Remote Sensing of Environment*, *330*, 114981.

---

## Author Comment (AC1)

Review of the manuscript "Origins, evolutions, and future directions of Landsat science products for advancing global inland water and coastal ocean observations", by Page et al.

**General comments**

I am quite confused about this paper. According to the authors, "The purpose of this paper is to communicate with aquatic scientists, satellite oceanographers, and the broader Earth observation community on the origins, requirements, challenges, successes, and future objectives for operationalizing global AR data products for Landsat satellite missions." However, I thought ESSD published papers introducing new datasets. The Landsat Level 2 provisional Aquatic Reflectance (AR) product has been available since 2020. I leave this to editorial decision, but I am not sure whether this journal is the right place for such a paper, which looks more like a review.

Author's response (red): We'd like to take this opportunity to thank you for your peer review. We think the purpose of the paper is properly stated. We cannot find on the ESSD web portal where the journal scope is restricted to new datasets. Although the Landsat Level 2 provisional Aquatic Reflectance (AR) product was released on-demand in 2020 by the USGS, there has not been a scientific peer review paper written describing this data product its validation, and plans for operational readiness. The authors are confident that ESSD is the right journal to publish the data description and its validation findings.

**Specific comments**

On Landsat 8/9 validation with in situ Rrs, I will be clear here. The gold standard for such a task is AERONET-OC data. AERONET-OC is the most precise, carefully calibrated data at disposal, with precisely characterized uncertainties, with all instruments within the network following the same procedures, software and calibration laboratories. Instead, GLORIA is pure noise, as already showed in Wei et al. (2025; Fig. 2), and once more shown in Figure 5 of this paper. GLORIA cannot tell anything about the satellite data being compared to, because the uncertainties in the in situ data are, most importantly, unknown. It is merged numbers from a diversity of research groups, most of which do not quite follow best practices on above-water radiometry, so to speak. The authors shall trust me, as I know virtually everybody who contributed to GLORIA. Regarding this, the sentence "Until validation campaigns are conducted on a routine basis, the combined GLORIA and AERONET-OC datasets offer a substantial validation pathway toward operational readiness" shall be removed. In summary, I encourage the authors to remove the comparison with GLORIA or leave them as a demonstration of its unsuitability for satellite validation.

Yes, we agree that the AERONET-OC data network is reference standard for evaluating the performance of aquatic and ocean colour algorithms and products. However, this data network is restricted to large bodies of water where the water column constituents, aquatic biology, and trophic state are not entirely representative of inland water bodies (e.g., small to medium sized) observed by medium spatial resolution satellites such as Landsat. We refrain from making quality judgements on the merits of GLORIA, but rather, treated it as one validation dataset that has previously been published and compiled as secondary datasets for meta-level analysis and

exploitation. We agree that compiled, community-contributed datasets are fraught with inconsistencies in the data collection, processing, and instrument performance. This fact only emphasizes the importance on consensus data collection protocols the growing need for such in situ radiometric datasets. We are comfortable with following revised assertion "Until in situ validation campaigns are conducted on a routine basis with standard operating procedures that are community-endorsed, the combined GLORIA and AERONET-OC datasets offer an interim validation pathway for assessing the operational readiness of aquatic and/or ocean colour processing algorithms and data products". We plan to retain the GLORIA dataset and analysis as part of this paper but take note of the reviewers suggestion to documents the limitations and have revised the text accordingly.

Although this may be a bit off-topic, it is of utmost importance that USGS works in the reprocessing of Landsat 5/7 for aquatic applications and, eventually, the release of a harmonized time series. That would accomplish the main mission of Landsat, which is the development of long-term series. I am aware of the difficulties due to the S/N, band configurations, etc., but any efforts to overcome any of these difficulties would be very impactful.

We appreciate the off-topic suggestion. The USGS is actively working to scope the reprocessing of the Landsat 5/7 record for aquatic applications. So far, we have found that instrument noise and limited atmospheric correction algorithm development for this historical data remain challenges to overcome, but we surely recognize the need for time series exploitation. A harmonized aquatic Landsat dataset with other complementary missions is also an important objective and will only come with time and resource commitments.

Line 36: what the authors call "mid-resolution" we actually call "high-resolution". Mid-resolution for us are sensors like MODIS or OLCI.

We agree slightly and have restated as 'medium resolution'. USGS defines high resolution as 10 meters or less. Terminology and semantics remain a challenge even today when describing remote sensing observations and physical quantities.

Line 60: "terrestrial land" sounds redundant.

We removed terrestrial and replaced with land change.

Lines 90-91: "characterize" sounds vague. What is being done, precisely?

We have substituted the word characterize with "monitor" for clarity.

Line 153: is the f\_b coefficient the "bidirectional reflectance correction"? I am not sure what is being discussed there, but the bidirectional reflectance correction is a complex process in the

aquatic environment, that involves a bio-optical model, IOP retrieval and some look-up-table indexing. See literature on the matter, especially recent articles, and clarify this concept.

Thank you for pointing this out. While the I2gen code within SeaDAS provides options for BRDF correction, this variable is unused in the processing. We have clarified this in the text by specifically including it in parentheses next to the variable definition under the equation.

Line 234: "difficulty of treating adjacency effects" Does it mean that I2gen treats the adjacency effect in some way? In this respect, please see a recent contribution by Castagna and Vanhellemont.

Correcting for adjacency effects during atmospheric correction remains an active area of algorithm research and development in remote sensing more broadly. The authors understand at the time of writing that solutions are beginning to emerge as suggested in the shared paper resource. At the time of USGS's SeaDAS algorithm integration into Landsat on-demand aquatic processing, the adjacency effects were ignored as has been common for many years and other peer reviewed communications. We recognize the importance of accounting for such effects and plan to implement such advancements upon research maturity and algorithm availability.

We have removed the sentence altogether as this is the first time adjacency effects was mentioned and is not relevant to the processing of this product.

Line 277: "community-driven" is more like "community-made"

We have modified "community-driven" to "community-made".

Lines 287-288: "allow the scientific community to characterize and validate Landsat AR algorithm outputs in near real time". I do not things so, because the AERONET-OC data is a reprocessed product, released some time after the actual measurement.

We agree with the reviewer. We meant to say that we are providing this Landsat curated aquatic validation dataset to enable the scientific community to rapidly assess the performance of Landsat AR algorithms and products prototypes during research and development. We are seeking to publish our validation methods in an open transparent manner to foster standardized for operational processing to aquatic analysis ready data.

We have removed the text "in near real time" to mitigate the false statement.

Section 4.2 "Validation methodology". I have to warn about the satellite vs. in situ data matchup time window. Same day is not close enough. That violates community agreed practices, see Concha et al. A few hours will be already too much, as the coasts are dynamic areas and often tidal.

We understand this matchup concern well. There are many scientific studies and papers that have either used this matchup criteria, or fact, have adopted a much looser threshold such as 1 day or more. We do agree that closer in time between in situ and satellite overpasses is ideal. To this end, we have revisited the validation methodology and restricted the time interval to be within +/- 3 hours, reducing the total matchups from 1,638 to 554 to follow common practice as seen in Concha et al. and other studies. Further, we included the median symmetric accuracy ( $\epsilon$ ) and signed symmetric bias ( $\beta$ ) as alternative metrics to determine algorithm performance that make it more communicable with other studies as seen in Wei et al., 2025 and the ACIX-agua excercise.

Equation (5): " $nLw_f/Q(\lambda)$ " that is quite a cumbersome notation. Please choose a better one. By that way, it actually means that the normalized water leaving radiance was corrected for bidirectional effects using Morel's method. Here, there are two issues:

- (1) Morel is a method conceptually developed for open oceanic waters. For coastal and inland waters, other methods should be used. See recent works.
- (2) If AERONET-OC data is corrected for bidirectional effects and Landsat data is not, as I suspect, that is an inconsistency. Both datasets shall be corrected, and using the same method.

We have corrected the scientific notation that properly reflects the updated conversion of nLw to Rrs without the f/Q(lambda) coefficient. Further, we have restricted AERONET-OC measurements to the same temporal window of +/-3 hours as with the GLORIA dataset.

**References**

Castagna, A., & Vanhellemont, Q. (2025). A generalized physics-based correction for adjacency effects. *Applied Optics*, *64*(10), 2719-2743.

Concha, J. A., Bracaglia, M., & Brando, V. E. (2021). Assessing the influence of different validation protocols on Ocean Colour match-up analyses. *Remote Sensing of Environment*, 259, 112415.

D'Alimonte, D., Kajiyama, T., Pitarch, J., Brando, V. E., Talone, M., Mazeran, C., ... & Gossn, J. I. (2025). Comparison of correction methods for bidirectional effects in ocean colour remote sensing. *Remote Sensing of Environment*, *321*, 114606.

Morel, A., Antoine, D., & Gentili, B. (2002). Bidirectional reflectance of oceanic waters: accounting for Raman emission and varying particle scattering phase function. *Applied Optics*, *41*(30), 6289-6306.

Wei, J., Wang, M., Jiang, L., Lee, Z., Kirby, R., Mikelsons, K., & Lin, G. (2025). Satellite observations of water transparency from VIIRS in global aquatic ecosystems. *Remote Sensing of Environment*, 330, 114981.

We have incorporated the Concha and Wei papers as suggested. We have excluded the Castagna, D'Alimonte and Morel papers as BRDF and adjacency corrections are not implemented in the processing of the Level 2 provisional AR products.

---

## Author Comment (AC2)

I think the paper is interesting and useful for the community. I suggest emphasizing the novelty of this study in the abstract/introduction, since there are similar previous studies. The authos must reorganized some part of the paper, infact some methodological part is present in the results and some discussion is in the results. Probably the authors must check well the units of the reflectances, in particular in the graphs.

Thank you for this observation. We have gone through and updated all figures that now have the proper reflectance unit labeling.

Some specific comments:

In lines 39-43: please add some references.

Please check all acronyms in the document; for example, Chlorophyll is written in three different ways.

We appreciate the consistency check. We have updated the acronyms and kept them consistent throughout the document.

In line 127: please clarify what "fr" is or, if it is an error, delete it.

Thank you for identifying this typo. It has been removed.

In line 293: please replace "sit" with "sites".

The sentence has been updated for clarification from "The limited number of inland platforms sit on" to "The limited number of inland platforms were installed on..."

In lines 319-321: I suggest expanding on the section on optical water types. In particular, mentioning the different classes that are then shown in Figure 8.

In line 358: please check the punctuation.

Lines 366-369 "The green (B3) ... (Pahlevan et al., 2021)", I suggest moving them and expand on them in the discussion.

We have corrected the sentence and relocated it to 391-395 in the results section.

In section 4 "Research methods": please add a couple of sentences describing the spectral resampling method that was performed for both in situ datasets. Also add the time window (in terms of hours) that was considered in the study between satellite observations and in situ measurements.

No spectral resampling of the in situ datasets was performed in this study. Instead, we selected the in situ Rrs values at wavelengths nearest to the Landsat OLI band center wavelengths. This approach was chosen to avoid introducing additional uncertainty

associated with spectral convolution or interpolation, which can be particularly sensitive to the shape of the in situ spectra and the accuracy of the sensor's relative spectral response (RSR) functions. While spectral resampling is often recommended for precise sensor comparisons, it can introduce its own uncertainties due to assumptions about spectral shape and instrument response. Given the relatively broad bandwidths of Landsat OLI and the high spectral resolution of our in situ data, the nearest-band approach was deemed sufficient for the objectives of this study. We have included this information in the methodology section of the manuscript at line 340.

We have also included the temporal window restriction and reasoning at line 315-317.

Before Figure 3, a couple of sentences introducing it are missing from the text. Also, check the reference axes, as they appear to be incorrect. "Reflectance"?

The axis labeling has been updated with the proper units for all figures.

For figures 5, and 6, it is not clear what unit is being used on the axes. Please clarify whether you are considering reflectance or Rrs. Make all graphs consistent with each other so that they are easier to interpret. Also, add the wavelengths (nanometers) to the graphs. For example, instead of using B1, I suggest to put 443 nm, and so on.

The axis labeling has been updated with the proper units for all figures. Additionally, we have changed the x-axis figures to include the OLI band center wavelength.

Lines 404-413: In this section, you discuss the protocols and methods used for the two in situ datasets. This section should be included in the methodology when you explain the datasets used in the work.

Thank you for this suggestion. We have moved this information to the methodology section for clarity.

Lines 413-414: This section should be included in the discussion. In addition, you could mention not only the uncertainties related to subjectivity in the choice of optical water types, but also briefly refer to some of the concepts mentioned above (404-413). For example, the fact that the AERONET dataset is consistent in its collection of measurements, unlike the GLORIA dataset. This could add more uncertainty to the analysis, and it would be better if you mentioned it in the discussion.

Lines 413-421: This part should be improved. It is a combination of results and discussion.

We have improved this portion of the section and have revised it in the updated results section.

I suggest adding a brief note at the beginning of the "Discussion" section regarding the results obtained in this work and the uncertainties associated with in situ measurements (as mentioned in points 13-14).

We have included a short summary in the discussion section regarding the results observed in the previous comment.

In line 531: Perhaps you are referring to criteria 3-4 and not 2-4. Please check.

Thank you for pointing out this typo. We are indeed referring to criteria 3-4, and have updated the sentence appropriately.